# Regulatory T Cell Depletion Using a CRISPR Fc-Optimized CD25 Antibody

**DOI:** 10.3390/ijms23158707

**Published:** 2022-08-05

**Authors:** Marit J. van Elsas, Johan M. S. van der Schoot, Alexander Bartels, Kas Steuten, Duco van Dalen, Zacharias Wijfjes, Carl G. Figdor, Thorbald van Hall, Sjoerd H. van der Burg, Martijn Verdoes, Ferenc A. Scheeren

**Affiliations:** 1Department of Medical Oncology, Oncode Institute, Leiden University Medical Centre (LUMC), Albinusdreef 2, 2333 ZA Leiden, The Netherlands; 2Department of Tumor Immunology, Radboud Institute for Molecular Life Sciences, Radboud University Medical Centre, Geert Grooteplein 26, 6525 GA Nijmegen, The Netherlands; 3Institute for Chemical Immunology, Geert Grooteplein Zuid 28, 6525 GA Nijmegen, The Netherlands; 4Department of Dermatology, Leiden University Medical Centre (LUMC), Albinusdreef 2, 2333 ZA Leiden, The Netherlands

**Keywords:** antibody, immunotherapy, CRISPR-HDR, Fc optimization, hybridoma

## Abstract

Regulatory T cells (T_regs_) are major drivers behind immunosuppressive mechanisms and present a major hurdle for cancer therapy. T_regs_ are characterized by a high expression of CD25, which is a potentially valuable target for T_reg_ depletion to alleviate immune suppression. The preclinical anti-CD25 (αCD25) antibody, clone PC-61, has met with modest anti-tumor activity due to its capacity to clear T_regs_ from the circulation and lymph nodes, but not those that reside in the tumor. The optimization of the Fc domain of this antibody clone has been shown to enhance the intratumoral T_reg_ depletion capacity. Here, we generated a stable cell line that produced optimized recombinant T_reg_-depleting antibodies. A genome engineering strategy in which CRISPR-Cas9 was combined with homology-directed repair (CRISPR-HDR) was utilized to optimize the Fc domain of the hybridoma PC-61 for effector functions by switching it from its original rat IgG1 to a mouse IgG2a isotype. In a syngeneic tumor mouse model, the resulting αCD25-m2a (mouse IgG2a isotype) antibody mediated the effective depletion of tumor-resident T_regs_, leading to a high effector T cell (T_eff_) to T_reg_ ratio. Moreover, a combination of αCD25-m2a and an αPD-L1 treatment augmented tumor eradication in mice, demonstrating the potential for αCD25 as a cancer immunotherapy.

## 1. Introduction

The presence of regulatory T cells (T_regs_) within a tumor contributes to a local environment in which effective anti-tumor immunity is actively dampened. In preclinical models, increasing numbers of T_regs_ support tumor manifestation whereas their depletion leads to a delay in tumor growth [1,2,3]. In the clinic, a low ratio of effector T cells (T_eff_) to T_regs_ (T_eff_/T_reg_) corresponds with a poor reaction to immunotherapy whereas a high T_eff_/T_reg_ ratio is associated with effective immunotherapy [4,5,6]. Targeted strategies to reduce the number of tumor-infiltrating T_regs_ or to diminish their immunosuppressive effects can offer significant benefits to clinical cancer treatments. In particular, their combination with T cell stimulatory immunotherapies such as the use of checkpoint inhibitors or the generation of tumor-specific T cell responses via vaccination regimens with tumor-specific antigens may improve therapy [7,8,9].

The recent clinical success of antibody-based immune checkpoint inhibitors in various malignancies has led to a surge in the development of novel immune modulators that alleviate immune suppression in the tumor microenvironment (TME) [10,11]. The therapeutic efficacy of these antibodies was first thought to be solely mediated by the Fab-mediated blockade of inhibitory signaling from T_regs_. However, recent preclinical and clinical evidence suggests that the Fc-mediated mechanism is a major determinant in leveraging therapy in checkpoint-targeted treatments [12,13,14]. The use of Fc-optimized αCTLA-4 resulted in superior tumor control in mouse models, and the presence of high-affinity fragment crystallizable (Fc) receptor (FcR) polymorphisms in patients treated with αCTLA-4 ipilimumab correlated with increased survival for these patients [14,15]. Therefore, the use of depletion-inducing backbones in antibodies targeting the surface proteins highly expressed on T_regs_ could be a valuable strategy to potentiate the efficacy of existing therapeutic antibodies and leverage cancer immunotherapy.

CD25, also known as the interleukin-2 high-affinity receptor alpha chain (IL-2Rα), represents a valuable target for the antibody-mediated depletion of T_regs_. This receptor is highly expressed in T_regs_ and, in combination with CD4 and forkhead box P3 (FoxP3) expression, is used for the identification of this immune-suppressive cell type [16]. CD25 has been used as a target for the development of antibody-based therapeutics in past studies; although preclinical and clinical studies have demonstrated that these antibodies can actively deplete T_regs_, their anti-tumor activity in vivo remained modest [3,17,18,19]. The clone PC-61 is a widely used anti-murine CD25 antibody that depletes T_regs_ in vitro, but is rather ineffective at depleting T_regs_ within established tumors [2,20]. This is attributed to the isotype of PC-61, rat IgG1, that exhibits a low preference for activating over-inhibiting Fc gamma receptors, which is a strong determinant for its therapeutic potential [21,22]. Others have demonstrated that exchanging the immune inert rat IgG1 Fc domain for the classical depleting mouse IgG2a Fc domain results in the augmented anti-tumor activity of PC-61 [23]. This αCD25-m2a version effectively depleted the intratumor T_regs_ in mouse models, increased the intratumoral T_eff_/T_reg_ ratios, and, in combination with an αPD-1 treatment, effectively eradicated tumors in vivo [23].

We are interested in αCD25-m2a as a sole T_reg_-depleting agent, which would allow us to study the underlying dynamics and timing of T_reg_ depletion and subsequent T_eff_ generation. We recently demonstrated efficient engineering of the immunoglobulin heavy chain locus of hybridomas by combining CRISPR-Cas9 (clustered regularly interspaced short palindromic repeats (CRISPR)-associated protein 9) with homology-directed repair (HDR) [24]. This allowed the rapid generation of stable cell lines that secreted isotype variants of the parental antibody without losing specificity for the original target. Here, we used this platform to obtain recombinant αCD25-m2a and evaluated its efficacy in combination with a programmed death ligand 1 (PD-L1) treatment for cancer immunotherapy.

## 2. Results

### 2.1. CRISPR-HDR Can Be Used to Optimize the PC-61 Hybridoma

To engineer the immunoglobulin locus of the rat IgG1 PC-61 hybridoma (Figure 1A) we utilized a guide RNA (gRNA), which directed the Cas9 nuclease to a protospacer adjacent motif (PAM) located in an intron upstream of the exon encoding the CH1 domain of the rat IgG1 (Appendix A). Additionally, we designed an HDR donor construct to resolve the Cas9-mediated double-stranded break and insert a synthetic exon consisting of an artificial splice acceptor, the murine IgG2a Fc domain, a sortag and his-tag, and gene elements to convert the blasticidin resistance (Figure 1B, Appendix A). After the co-transfection of the PC-61 hybridomas, we applied a selection pressure with blasticidin to enrich the hybridoma clones, which successfully integrated the donor construct. After one week, we confirmed the secretion of proteins with the his-tag in the supernatant of the selected hybridoma population and performed limiting dilution to obtain monoclonal cell lines. His-tag-positive monoclonal supernatants, as determined by a dot-blot (Figure 1C), were exposed to CD25^+^ target cells in combination with secondary antibodies against murine IgG2a or rat IgG1 to select a recombinant hybridoma that exclusively expressed the CD25-binding antibodies of the murine IgG2a isotype (Figure 1D). The reliable quantification of the insertion was not possible because only hybridomas with a successful insertion survived. However, this approach had a success rate of 100%, meaning that every genomic engineering experiment using a single specific HDR donor construct resulted in multiple correctly engineered hybridoma clones after the selection (Figure 1C,D). Based on the exclusive and high expression of murine IgG2a, we selected the recombinant hybridoma clone PC-61_B2 for the production of αCD25-m2a. After the cultivation of PC-61_B2, we isolated αCD25-m2a from the supernatant using his-tag affinity purification. We subsequently cleaved off the his-tag via sortase (srt)-mediated hydrolysis [25] (Figure 1E,F) to circumvent the potential his-tag-associated liver retention of the antibodies [26]. Notably, within a fraction of αCD25-m2a, the use of a sortase-mediated cleavage of the his-tag resulted in the intramolecular covalent linkage of the heavy chains (Appendix A), likely through the nucleophilic addition of the ε-amino group of the lysine present near the c-terminus to the sortase-activated c-terminus of the adjacent heavy chain [27].

To evaluate the T_reg_ depletion efficiency of our αCD25-m2a, we tested the antibody in the syngeneic MC38 colon adenocarcinoma in vivo tumor model (Figure 2A). As the activity of the original antibody αCD25-r1 has been well-studied by others, we opted not to use αCD25-r1, but only an isotype control (IgG2a) reference group for ethical reasons. We administered 200 µg of αCD25-m2a on days 5 and 7 after inoculating mice with MC38 and evaluated the T cell populations within the blood, draining the lymph node and tumor on day 9. The treatment with αCD25-m2a reduced the fraction of CD25-expressing cells in the CD4^+^ T cell compartment (Figure 2B,C), decreased the percentage of CD4^+^FoxP3^+^ T_regs_ (Figure 2C), and increased the T_eff_/T_reg_ ratios (Figure 2C) in the blood, draining the lymph nodes and tumor. The depletion of CD25^+^FoxP3^+^ CD4^+^ T cells was nearly complete in all compartments (Appendix A). Importantly, the depletion of effector CD4^+^ and CD8^+^ T cells was less dramatic compared with the depletion of T_regs_ (Appendix A). Only the percentage of intratumoral CD8^+^ T_effs_ (CD8^+^CD25^+^) and CD4^+^ T_effs_ in the tumor and drained lymph nodes (CD25^+^FoxP3^−^), but not that of the circulating T_effs_, was significantly decreased (Appendix A).

### 2.2. CD25-m2a Treatment Synergizes with an αPD-L1 Treatment

Having established that αCD25-m2a efficiently depleted tumor-infiltrating T_regs_, we wanted to explore its therapeutic efficacy in controlling tumor growth. As antibodies targeting the PD-1/PD-L1 axis increase the T_eff_ function and anti-tumor activity, we hypothesized that αCD25 would synergize with αPD-L1 in a combinational therapy. Therefore, we inoculated mice with MC38 and injected 200 µg αCD25-m2a on day 5 and day 7; injected 200 µg of the αPD-L1 clone 10F.9G2 (rat IgG2b) on day 5, day 8, and day 11; or combined the two treatment regimens and measured the tumor growth over 60 days (Figure 3A). Although a monotherapy with αCD25-m2a or an αPD-L1 treatment resulted in the respective survival of 2/10 and 2/9 mice at the endpoint of the study, the tumor growth in most mice was not delayed and was comparable with the control group (Figure 3B,C). Of note, the mice that responded to the αCD25-m2a treatment experienced tumor regression as late as 30 days after the tumor inoculation, indicating that the CD25-mediated depletion of T_regs_ had a delayed impact on the tumor growth. By combining αCD25-m2a and the αPD-L1 treatment, the number of mice responding to therapy increased (8/9), resulting in a significant improvement in the survival and response rates (6/9) at the study termination compared with the other treatments (*p* < 0.05, Figure 3C,D). A delayed response was also observed that was similar to the responders in the αCD25-m2a monotherapy group. Taken together, these data demonstrate that the recombinant αCD25-m2a antibody stably produced by our CRISPR-HDR-engineered PC-61 hybridoma was able to efficiently deplete T_regs_ from the TME and synergized with the αPD-L1 treatment of the MC38 mouse colon adenocarcinoma.

### 2.3. Continued T_reg_ Depletion Does Not Affect Therapeutic Efficacy

The expression of CD25 was not limited to T_regs_; it was also present on activated CD4^+^ and CD8^+^ T cells. The treatment with CD25-m2a resulted in a decrease in the intratumorally activated CD4^+^ and CD8^+^ T_eff_ cells (Appendix A). Despite this decrease, the observed durable tumor control led us to hypothesize that this activated CD4^+^ and CD8^+^ T_eff_ cell population was more rapidly renewed compared with the T_reg_ population, resulting in a beneficial T_eff_/T_reg_ ratio. To test this hypothesis, we treated MC38-tumor-bearing mice with 200 µg αCD25-m2a on day 5 and day 7 (short-term) or with 200 µg αCD25-m2a on days 5, 7, 11, 16, 20, and 25 (continued) in combination with 200µg αPD-L1 (Figure 4A). Surprisingly, a continued T_reg_ depletion did not affect the tumor control; 4/8 mice had a durable tumor control compared with 6/10 in the short-term depletion group (*p* = 0.63, Figure 4B–D). This suggested a low replenishment rate of T_regs_ in the tumor as no benefit to the tumor control or survival could be observed. Moreover, as no negative effect on the survival was seen following the continued αCD25-m2a treatment, we concluded that a continued αCD25-m2a treatment did not negatively affect the anti-tumor immune response.

## 3. Discussion

Here, we described the generation of a stable cell line continuously producing recombinant Fc-optimized anti-murine CD25 antibody PC-61 with an enhanced immunotherapeutic activity for cancer treatment. For this purpose we performed CRISPR-HDR engineering of the parental hybridoma and exchanged its immune inert rat IgG1 Fc domain for the classical depleting murine IgG2a backbone (Figure 1A). Genetic engineering was effective as we were able to identify multiple cell lines that secreted CD25-specific antibodies of the murine IgG2a isotype with the designed chemoenzymatic- and his-tag. After isolation and his-tag removal, we performed tumor-challenging studies with αCD25-m2a and demonstrated that the Fc-optimized antibody could efficiently deplete T_regs_ from the TME (Figure 2). This was in contrast to the original PC-61 clone with the rat IgG1 isotype that is unable to clear tumor-infiltrating T_regs_, and demonstrated a modest anti-tumor activity in vivo in combination with other therapies [28,29]. Moreover, the αCD25-m2a treatment synergized with the αPD-L1 treatment, leading to the effective elimination of tumors and a significant increase in survival compared with the use of either treatment alone (Figure 3) [2,20]. Together, these results demonstrate that CRISPR-HDR can be used to repurpose hybridomas and their antibodies for cancer immunotherapy.

In addition to increasing therapy efficacy, the reduction of treatment-associated toxicities has become a priority for the development of novel immunotherapeutic treatments. In the clinic, ipilimumab is one of the most commonly used immunotherapies; its therapeutic efficacy has been attributed not only to the blocking of the CTLA-4 function, but also to the depletion of T_regs_ [30]. However, ipilimumab is also associated with the development of severe immune-related toxicities, which may be attributed to the activation of T_eff_-recognizing self-antigens [31,32,33,34]. In comparison with the αCTLA-4 treatment, we hypothesized that the αCD25 treatment could offer a larger therapeutic window because CD25 lacks an immune modulatory capacity compared with CTLA-4. Therefore, our results warrant a further investigation of αCD25 as a potential therapeutic.

In this study, we showed that the αCD25-m2a treatment synergized with the αPD-L1 treatment, resulting in superior tumor control and increased survival in vivo (Figure 3). Although the αCD25-m2a treatment alleviated tumor suppression by T_reg_ depletion, the αPD-L1 treatment is known to block PD-L1 ligation to PD-1 on activated T cells, which circumvents the impairment of the T cell function [35,36]. Therefore, the PD-L1 blockade promotes the activity of the T_eff_ compartment within the TME. However, this particular αPD-L1 clone, 10F.9G2, has multiple mechanisms of action and can, besides the blockade of the PD-1/PD-L1 axis, directly deplete tumor cells as well as certain myeloid populations within the tumor [37]. Therefore, the extrapolation of which mechanism contributes to the synergistic effect is difficult and will be explored in future work by the characterization of T cell and myeloid populations at different timepoints during and after treatment.

In the past, CD25 received much attention as a potential target for cancer immunotherapy as it is used for the identification and isolation of T_regs_. This lack of T_reg_-depleting efficacy in the tumor of the PC-61 antibody can be attributed to the rat IgG1 isotype and differential FcγR expression of effector cells in the circulation, blood and tumor. Specifically, FcγR engagement of rat IgG1 is limited to inhibiting FcγRIIb and activating FcγRIII and can, therefore, only recruit effector cells for T_reg_ depletion when these express low levels of FcγRIIb [38]. In the circulation and lymph nodes, macrophages express low levels of FcγRIIb and can, therefore, be recruited for T_reg_ depletion [23]. In contrast, macrophages in the TME express high levels of FcγRIIb, impairing macrophage activation and the depletion of T_regs_ [23,39]. The use of the mIgG2a isotype backbone in PC-61, which interacts beside FcγRIIb and FcγRIII with activating FcγRI and FcγRIV [40], can overcome this inhibitory signaling of FcγRIIb and may license tumor-associated macrophages for T_reg_ depletion. These observations underline the importance of evaluating different antibody formats early in development to select an optimal design that augments the therapeutic potential, as demonstrated by others [41,42].

The depletion of T_regs_ by CD25-targeting antibodies has already been tested in a clinical trial in combination with a dendritic cell (DC) cancer vaccine [19]. Despite a successful clearance of T_regs_ from the circulation of these patients, no effect on progression-free survival could be observed compared with the DC vaccine single treatment. This lack of response has been attributed to the αCD25-mediated depletion of activated CD25^+^ effector T cells, required for DC vaccine-induced tumor control. In line with these findings, we also found that despite the favorable T_eff_/T_reg_ ratio in the tumor, the number of activated T_effs_ also decreased, as determined by the percentage of CD25^+^ cells in the CD8^+^ T cell compartment on day 9 (Appendix A). The reduction of activated T_effs_ within the tumor offered an explanation for the delayed response to the αCD25-m2a/αPD-L1 treatment. Surprisingly, even after the continued depletion of CD25^+^ T cells by CD25-m2a, a durable tumor control was observed (Figure 4). Therefore, we established that this subset of activated CD25^+^CD8^+^ T cells that was depleted by CD25-m2a was not crucial for anti-tumor T cell activity. Sophisticated efforts to circumvent this undesirable depletion of CD25^+^ T_eff_ cells by αCD25-targeting antibodies have been undertaken, resulting in the development of a non-IL-2 blocking αCD25 antibody [43]. By resolving the variable region of the anti-CD25 clone 7D4 and cloning it into a mouse IgG2a backbone, researchers have managed to generate a T_reg_-depleting antibody whilst preserving the T_eff_ function. Future research will determine whether the use of such an antibody could also improve therapeutic efficacy in the clinic.

To negate any possible detrimental in vivo effects of the c-terminal his-tag on Fc-optimized αCD25-m2a antibodies, we performed a sortase-mediated hydrolysis of the heavy chain. Although the his-tag was successfully removed, we observed an additional protein band of approximately 120 kDa in size as determined by reducing SDS-PAGE (Appendix A). Given that mass spectrometry measurements and non-reducing SDS-PAGE confirmed the final products to be monomeric and of the expected size, we postulated that the sortase-mediated reaction could lead to covalent inter-heavy chain linkage within a single antibody molecule. It is likely that the acyl-enzyme intermediate that was formed during the reaction was not resolved via hydrolysis, but via a nucleophilic attack from the ε-amino side chain of a lysine of the opposing heavy chain in close proximity to the c-terminus, linking the chains together [27]. Although the formation of these bonds did not influence the antibody performance, it could be prevented by providing an excess of nucleophilic peptides during the sortase-mediated reaction, or by simply removing the protein tags from the original HDR design.

Here, we adapted our previously developed CRISPR-HDR engineering approach [24] to modify several rat IgG2a hybridomas and obtain antibodies in the format and isotype of choice to accommodate the engineering of the rat IgG1 locus. The efficient generation of daughter cell lines secreting αCD25-m2a antibodies underlines the versatility of this approach as it can be adapted and applied to virtually any hybridoma. Such an approach can be advantageous over conventional transient techniques because: (i) no knowledge regarding Ig DNA sequence hybridomas is needed as only the isotype needs to be determined; (ii) a stable continuous source of an antibody product is preferred over transient systems; and (iii) expertise in and funds to use transient expression systems are not available.

## 4. Materials and Methods

**CRISPR-HDR engineering:** The genomic sequence of the IgH locus of rat IgG1 was identified via Ensembl rate genome build Rnor_6.0 for gRNA and donor construct design. The donor construct was ordered as a synthetic gene from IDT and cloned into a vector via traditional cloning techniques. The gRNAs, including gRNA-R1 (CATCCTCCATTTATACAGCC) used for this study, were designed with the CRISPR tool from the Zhang laboratory and cloned into the pX330-U6-Chimeric_BB-CBh-hSpCas9 vector, which was obtained as a kind gift from F. Zhang (Addgene ref: 42230) [44]. PC-61 hybridomas were transfected with the donor construct and gRNA-R1 and cultivated for 10 days under blasticidin pressure. The resulting blasticidin-resistant population was diluted in 96-well plates to 0.3 cells/well to obtain the monoclonal cell lines. The plates were cultured until the wells could be observed with high cell concentrations. From each monoclonal cell line, 3 µL supernatant was spotted onto a nitrocellulose membrane and allowed to dry at RT for 1 h. Subsequently, the nitrocellulose membrane was stained with rabbit antibody against his-tag (ab137839, Abcam, Cambridge, UK) and IRDye 800CW goat antibody against rabbit (926-32211, LI-COR, Bad Homburg, Germany) to select the resistant modified hybridomas that expressed his-tagged antibodies. His-tag-positive supernatants were used to stain CD3- and CD28-stimulated splenocytes from Bl6 mice and were subsequently screened with Rat anti-Mouse IgG2a-PE (12-4210-82, Thermo Fisher Scientific, Waltham, MA, USA) and Mouse anti-Rat IgG1-APC (17-4812-82, Thermo Fisher Scientific, Waltham, MA, USA) to identify the daughter cell lines that exclusively secreted the murine IgG2a that binds to CD25. A detailed description of CRISPR-HDR editing of hybridomas can be found elsewhere [24].

**Antibody production and isolation:** For αPC-61-m2a, 20 million PC-61_B2 cells were seeded in CELLine Bioreactor Flasks (900-05, Argos, Vernon Hills, IL, USA) and cultivated for 1 week. The antibody-containing supernatants were separated from the cells via centrifugation (90× *g*, 5 min), filtered through a 20 µm filter, and supplemented with 10 mM imidazole (I2399, Sigma-Aldrich, Saint Louis, MO, USA). Subsequently, the hybridoma supernatant was run over an Ni-NTA column (30210, Qiagen, Hilden, Germany) and washed with 10 column volumes of PBS supplemented with 10 mM imidazole before the antibody elution with PBS supplemented with 250 mM imidazole. The buffer exchange to PBS was performed via ultracentrifugation with Amicon Ultra-15 centrifugal filter units (Z717185, Sigma-Aldrich, Saint Louis, MO, USA). The WT antibodies from PC-61 were obtained by cultivating parental hybridomas using a CD hybridoma medium (11279023, Thermo Fisher Scientific, Waltham, MA, USA) supplemented with 2 mM ultraglutamine, 1 × AA, and 50 µM 2-ME. PC-61 was purified from the medium using a Protein G GraviTrap (28-9852-55, Sigma-Aldrich, Saint Louis, MO, USA) according to the manufacturer’s protocol. The antibody concentration was determined via absorption at 280 nm with an extinction coefficient of 1.4 using a Nanodrop 2000 (Thermo Fisher Scientific, Waltham, MA, USA). The protein purity was assessed by reducing SDS-PAGE using a SYPRO Ruby Protein Gel Stain (S12000, Thermo Fisher Scientific, Waltham, MA, USA).

**Sortase-mediated hydrolysis:** For his-tag removal, αCD25-m2a was dialyzed into a sortase buffer (150 mM tris, 50 mM NaCl, and 10 mM CaCl_2_; pH 7.5) via ultracentrifugation. The sortase-mediated reaction was carried out with 15 mg of αCD25-m2a with 0.5 equivalents of trimutant sortase A Δ59 (3M srt) [45] in a volume of 1.5 mL overnight at RT. Subsequently, the reaction volume was increased to 5 mL and adjusted to a concentration of 10 mM imidazole before it being incubated with 2 mL Ni NTA resin to remove the 3M srt, unreacted αCD25-m2a, and cleaved his-tag. The column was washed twice with PBS supplemented with 10 mM imidazole, after which the flow-through containing the hydrolyzed product was collected and dialyzed against PBS. The concentration was determined via absorption at 280 nm with an extinction coefficient of 1.4 using a Nanodrop 2000 (Thermo Fisher Scientific, Waltham, MA, USA) and analyzed on reducing and non-reducing SDS-PAGE to assess the purity. The his-tag removal of αCD25-m2a was assessed by a Western blot stained with an anti-his-tag rabbit antibody (ab213204, Abcam, Cambridge, UK) visualized by anti-rabbit IRDye800 (926-32211, LI-COR, Lincoln, NE, USA) and anti-mouse IRDye600 (9260-32220, LI-COR, Lincoln, NE, USA).

**Tumor experiments:** For the assessment of the T_reg_ depletion, female C57/BL6 mice were subcutaneously injected into their right flank with 5 × 10^5^ MC38 cells resuspended in 200 μL PBS supplemented with 0.1% BSA. To 2 cohorts of 5 mice, 200 µg of αCD25-m2a or an isotype control (IgG2a, InVivoPlus BioXCell, Lebanon, NH, USA) were intraperitoneally administered on days 5 and 7. The tumor volume was measured on days 5, 7, and 9, after which the mice were euthanized. The blood, lymph nodes, and tumors were harvested and processed for phenotyping using flow cytometry on the BD LSRFortessa flow cytometer (BD Bioscience, Franklin Lakes, NJ, USA) and analyzed using the FlowJo software (v10, Treestar, Ashland, OR, USA). The anti-mouse antibodies used for the flow cytometry were anti-CD4-BV711 (563050, BD Bioscience, Franklin Lakes, NJ, USA), anti-CD25-FITC (553072, BD Bioscience, Franklin Lakes, NJ, USA), anti-CD8-PerCP-Cy5.5 (551162, BD Bioscience, Franklin Lakes, NJ, USA), anti-FoxP3-PE (560046, BD Bioscience, Franklin Lakes, NJ, USA), anti-CD3-APC (100235, Biolegend, San Diego, CA, USA), and anti-CD45-aF700 (103127, Biolegend, San Diego, CA, USA). For the tumor challenge, 9 cohorts of 10 female C57/BL6 mice each were subcutaneously injected into their right and left flank with 5 × 10^5^ MC38 cells resuspended in 200 μL PBS supplemented with 0.1% BSA on day 0. All in vivo experiments were performed with 8–10 animals per group and performed once. For the control and short-term αCD25-m2a cohort, the mice received the same treatment as described above. For the continued αCD25-m2a cohort, additional treatments with 200 μg of αCD25-m2a were administered on days 11, 16, 20, and 25. For all the αPD-L1 treatment cohorts, the mice were injected on day 7 with an isotype control in 100 µL PBS and injected on days 5, 8, and 11 with 200 µg of αPD-L1 (BE0101, BioXcell, Lebanon, NH, USA). The tumor volume was determined every other day and the mice were euthanized when the tumor volume reached 1500 mm^3^. The mice were maintained under specific pathogen-free conditions. 

**Data Analysis:** Flow cytometry data were analyzed with FlowJo v10 (Tree Star, Ashland, OR, USA). Statistical analyses were conducted using Prism 8 (GraphPad Software, San Diego, CA, USA) and *p*-values were calculated using independent Student’s *t*-tests (ns *p* > 0.05; * *p* ≤ 0.05; ** *p* ≤ 0.01; *** *p* ≤ 0.001). Kaplan–Meier curves were analyzed with the log-rank Mantel–Cox test.

## Figures and Tables

**Figure 1 ijms-23-08707-f001:**
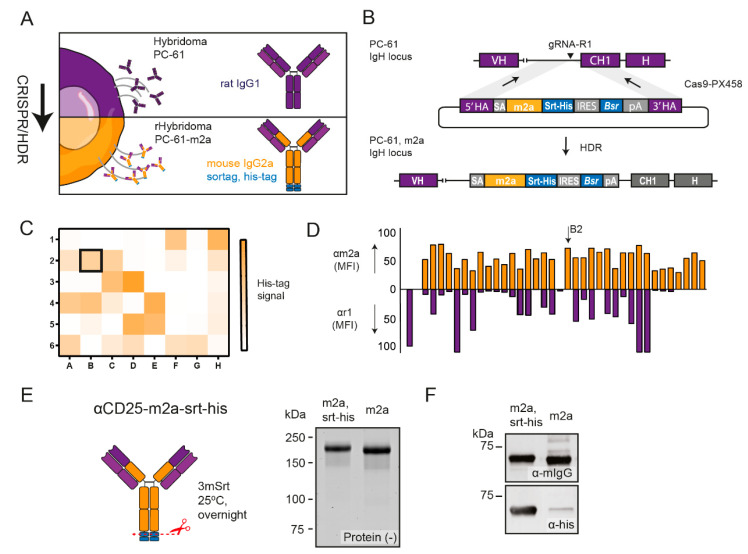
CRISPR-HDR-mediated Fc optimization of PC-61 hybridoma. (**A**) Schematic of CRISPR-HDR engineering to tune the expression of PC-61 hybridoma from rat IgG1 to mouse IgG2a. (**B**) The IgH locus of PC-61 was targeted with gRNA-R1 to create a double-stranded break that could be repaired through homology with the donor construct that inserted the murine IgG2a Fc domain with c-terminal sortag and his-tag (srt-his) and gene elements for blasticidin resistance. (**C**) Supernatant HDR and gRNA-R1-transfected cells that were resistant to blasticidin were taken and screened for secretion of his-tag-positive antibodies via dot-blot. (**D**) His-tag-positive supernatants were used to stain activated splenocytes in combination with secondary antibodies against mouse IgG2a (yellow) and rat IgG1 (violet) to select the engineered hybridoma that exclusively secreted m2a antibodies targeting CD25. The selected PC-61-m2a clone for antibody production is indicated (B2). (**E**) His-tag removal of PC-61-m2a via sortase-mediated hydrolysis with non-reducing SDS-PAGE and immunoblot. (**F**) αCD25-m2a efficiently depleted T_regs_, resulting in an increased T_eff_/T_reg_ ratio.

**Figure 2 ijms-23-08707-f002:**
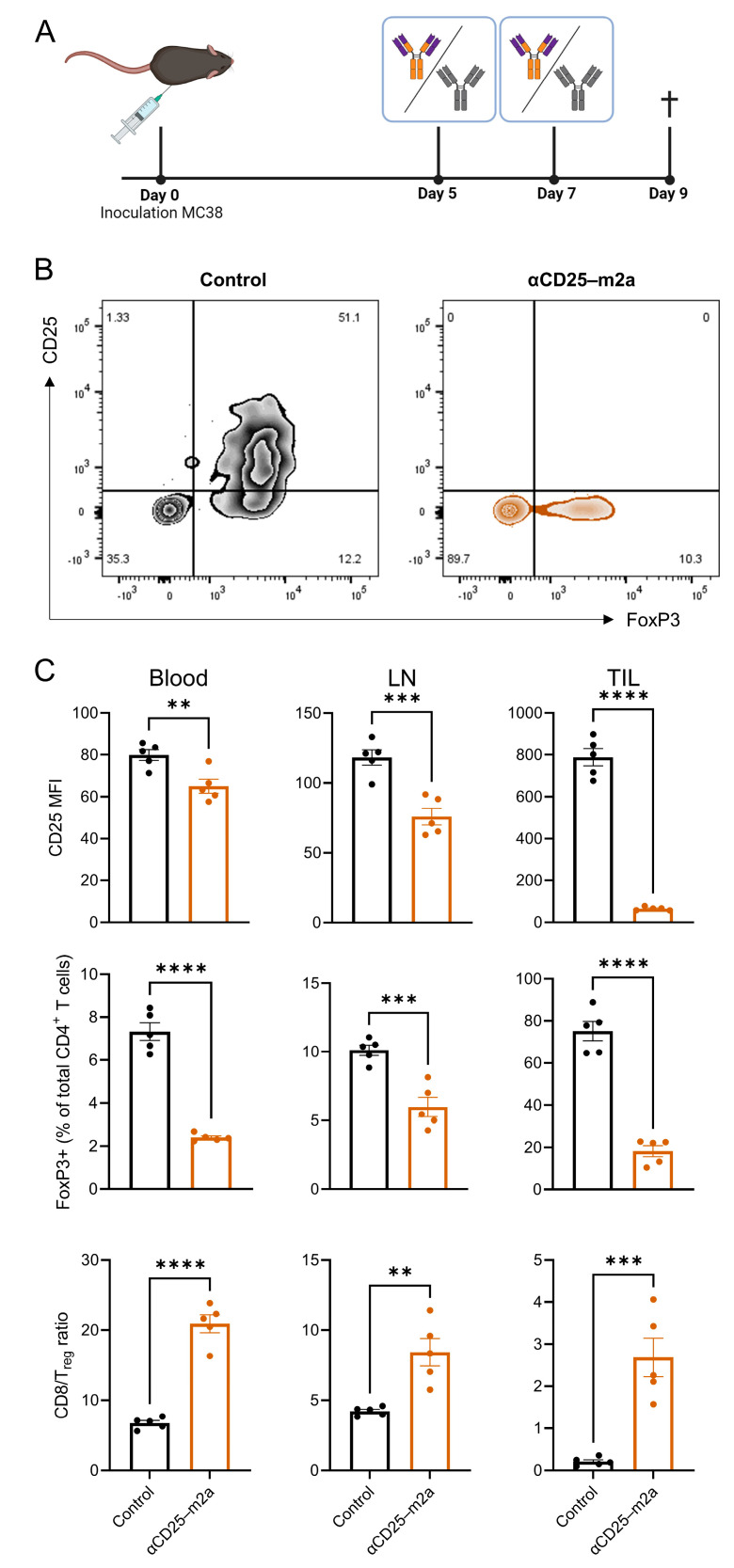
Effective depletion of T_regs_ in blood, lymph nodes, and tumors via Fc-optimized αCD25-m2a. (**A**) Schematic of tumor challenge in C57/Bl6 mice. On day 0, mice were subcutaneously inoculated with MC38 and on days 5 and 7 were treated with either αCD25-m2a or an isotype control. Blood, lymph nodes, and tumors were harvested on day 9. (**B**) Representative plots showing expression of CD25 and FoxP3 in CD3^+^CD4^+^ T cells isolated from tumors. Numbers show percentage of cells in each quadrant. (**C**) MFI of CD25 in T_regs_, percentage of T_reg_ cells in CD3^+^CD4^+^ T cells, and T_eff_/T_reg_ cell ratios in blood, lymph nodes, and tumors (*n* = 5, statistical difference determined via independent Student’s *t*-test: ** *p* ≤ 0.01; *** *p* ≤ 0.001; **** *p* < 0.0001).

**Figure 3 ijms-23-08707-f003:**
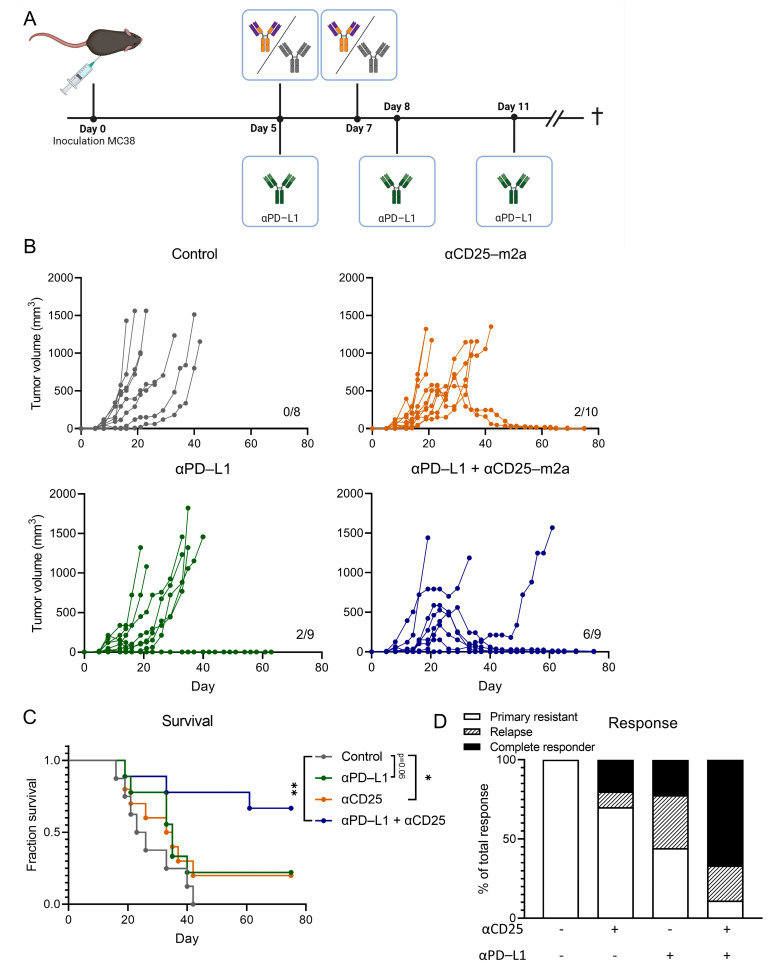
CD25-m2a treatment synergizes with αPD-L1 treatment. (**A**) Schematic of treatment schedule of C57/BL6 mice inoculated with MC38. On day 0, mice were subcutaneously inoculated with MC38 and subsequently treated on days 5 and 7 with αCD25-m2a or an isotype control, or treated on days 5, 8, and 11 with αPD-L1, or received both αCD25-m2a and the αPD-L1 treatment. Mice were euthanized when tumors reached a volume of 1500 mm^3^. (**B**) Growth curves of MC38 tumors. Number of tumor-free survivors is shown in each graph. (**C**) Survival curves and (**D**) response rates of mice shown in (**B**) (*n* = 8–10, statistical difference determined via log-rank test). Significance is shown as * = *p* < 0.05, ** = *p* < 0.01.

**Figure 4 ijms-23-08707-f004:**
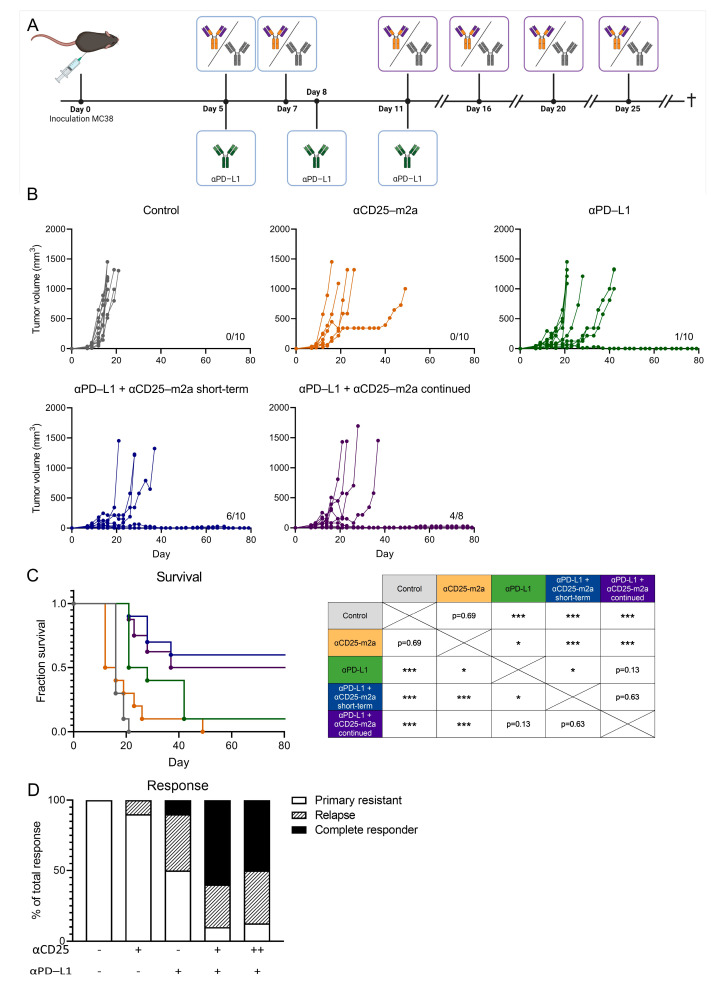
Continued T_reg_ depletion does not affect therapeutic efficacy. (**A**) Schematic of treatment schedule of C57/BL6 mice inoculated with MC38. On day 0, mice were subcutaneously inoculated with MC38 and subsequently treated on days 5 and 7 with αCD25-m2a (short-term) or days 5, 7, 11, 16, 20, and 25 (continued) or an isotype control, or treated on days 5, 8, and 11 with αPD-L1, or received both αCD25-m2a and the αPD-L1 treatment. Mice were euthanized when tumors reached a volume of 1500 mm^3^. (**B**) Growth curves of MC38 tumors. Number of tumor-free survivors is shown in each graph. (**C**) Survival curves including significance table and (**D**) response rates of mice shown in (**B**) (*n* = 8–10, statistical difference determined via log-rank test). Significance is shown as * = *p* < 0.05, and *** = *p* < 0.001.

## Data Availability

The data needed to evaluate the conclusions are present in the paper and/or the Appendix A. The described plasmids used in this study are deposited in the plasmid repository of Addgene (www.addgene.org/, accessed on 1 August 2022). Additional data and materials related to this paper may be requested from the authors.

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
