# Peer review of "Regulatory T Cell Depletion Using a CRISPR Fc-Optimized CD25 Antibody"

_ijms, 2022, doi:10.3390/ijms23158707_

Round 1
Reviewer 1 Report
Revision of the article :
“Regulatory T cell depletion using a CRISPR Fc-optimized CD25 antibody”.
The articles describes the optimization a CD25 antibody (PC-61), used for the depletion of Treg, by modification of the Fc from rat IgG1 to a mouse IgG2a. This modification should improve the anti-tumor activity of the antibody. Indeed, the authors could show that the newly generated αCD25-m2a antibody lead to the elimination of Treg, known to contribute to tumor progression and to an increased in Teff/Treg ratio. Furthermore, in tumor mice model, the αCD25-m2a antibody in combination with PD-L1 treatment leads to a better eradication of the tumor and prolonged survival.
I consider that article has interesting data and I would recommend its publication after the revisions below.
Major considerations.
- Line 26: Please indicate that m2a is mouse IgG2a Isotype.
- In the point 2.1.: Can you please provide results/indicate the efficiency of CRISPR-HDR in inserting the insert the murine IgG2a Fc domain .
- Line 99 and Figure 1: In the figure 1 it is indicated B2, so it is a B1 or B2. Please indicate the number/code for the other clones. From the Figure 1D, other clones could be consider positive for the IgG2a while negative for the IgG1 (I observed at least two), can you please comment on it!
- Line 101: Please include the abbreviation of sortase (srt) at it used several time below.
- Line 125: Can you please indicate which isotype control was used ( Was it produced similarly to the the αCD25-m2a antibody?)
- Line 184-185: Can you please comment, that the continued αCD25-m2a treatment also does not provide benefits in tumor control?
In the Discussion: As a suggestion, I would include figure number during the discussion to help the reader to follow the discussion session.
- Line 218: I would suggest to replace “believe” by “hypothesize”.
- Line 258-259: I would not conclude it, as it was not possible to observed a 100% survival. I would suggest that the depletion of this population seems not be crucial for the anti-tumor T cell activity. Further studies would be required to evaluate the importance of this population for anti-tumor immune response.
- Line 308: Please clarify, B1 or B2?
Supplementary Figure 2:
Please and for comparison show the Non- and reducing SDS-page from the same weight, i.e. from 250 kDA to 20 KDa for both.
Also indicate the units of the values for Maldi.
Please show the graph of MS highlighting the picks for αCD25-m2a and αCD25-m2a-srt-his.
Other comments:
Although it is optional, I would suggest to include a paragraph with the conclusion.
Along the text, please modify “in vivo” to italic.
For the Figure 3, Figure 4 and Supplementary Figure 3: Please indicate the statistical test used and p values legend (as done in Figure 2).
Please also indicate for the animal experimentation, how many time the assay was done.
Author Response
Reviewer 1
Revision of the article :“Regulatory T cell depletion using a CRISPR Fc-optimized CD25 antibody”.
The articles describes the optimization a CD25 antibody (PC-61), used for the depletion of Treg, by modification of the Fc from rat IgG1 to a mouse IgG2a. This modification should improve the anti-tumor activity of the antibody. Indeed, the authors could show that the newly generated αCD25-m2a antibody lead to the elimination of Treg, known to contribute to tumor progression and to an increased in Teff/Treg ratio. Furthermore, in tumor mice model, the αCD25-m2a antibody in combination with PD-L1 treatment leads to a better eradication of the tumor and prolonged survival.
I consider that article has interesting data and I would recommend its publication after the revisions below.
Major considerations.
- Line 26:Please indicate that m2a is mouse IgG2a Isotype.
The authors have added it to the manuscript (line 26)
- In the point 2.1.:Can you please provide results/indicate the efficiency of CRISPR-HDR in inserting the insert the murine IgG2a Fc domain.
The method is described in (van der Schoot et al. 2019) and the selection method does not allow for reliable quantification of the insertion as only hybridomas with successful insertion will survive. However: The approach has a success rate of 100%, meaning that every genomic engineering experiment using a single specific HDR donor construct resulted in multiple correctly engineered hybridoma clones after selection as seen in figure 1C and 1D. The authors have added this to the manuscript (lines 99-103).
- Line 99 and Figure 1:In the figure 1 it is indicated B2, so it is a B1 or B2. Please indicate the number/code for the other clones. From the Figure 1D, other clones could be consider positive for the IgG2a while negative for the IgG1 (I observed at least two), can you please comment on it!
- The correct clone was B2, the authors have corrected this throughout the manuscript.
- After clonal selection and isotyping of secreted antibody of the clones, we chose clone B2 from multiple clones that expressed IgG2a and lacked rIgG1 expression, as this was the clone with the highest expression of IgG2a. This is also mentioned in the manuscript “Based on the exclusive and high expression of murine IgG2a, we selected the recombinant hybridoma clone PC-61_B2 for production of αCD25-m2a.” (lines 103-104).
- Line 101:Please include the abbreviation of sortase (srt) at it used several time below.
The authors have added this (line 106).
- Line 125:Can you please indicate which isotype control was used ( Was it produced similarly to the the αCD25-m2a antibody?)
The isotype used was the commercially available IgG2a InVivoPlus antibody from BioXCell, clone C1.18.4. The authors have added this to the manuscript (line 130, 365).
- Line 184-185:Can you please comment, that the continued αCD25-m2a treatment also does not provide benefits in tumor control?
The lack of additional benefit from continued aCD25-m2a treatment over short-term depletion suggests a low replenishment rate of Tregs in the tumor. We hypothesize that short-term treatment is sufficient to deplete intratumoral Tregs for a substantial time and that continued depletion therefore does not result in survival benefit. The authors have added this to the manuscript (line 193-196).
In the discussion:
- As a suggestion, I would include figure number during the discussion to help the reader to follow the discussion session.
We agree with the reviewer and we have added the figure numbers throughout the discussion section of the manuscript.
- Line 218:I would suggest to replace “believe” by “hypothesize”.
The authors have replaced “believe” by “hypothesize” (line 232).
- Line 258-259:I would not conclude it, as it was not possible to observed a 100% survival. I would suggest that the depletion of this population seems not be crucial for the anti-tumor T cell activity. Further studies would be required to evaluate the importance of this population for anti-tumor immune response.
The authors agree with the reviewer and have adjusted the manuscript according to the reviewers’ suggestion (line 277-278).
- Line 308:Please clarify, B1 or B2?
B2, corrected throughout the manuscript.
Supplementary Figure 2:
- Please and for comparison show the Non- and reducing SDS-page from the same weight, i.e. from 250 kDA to 20 KDa for both.
For reliable protein analysis in the optimal molecular weight range the non- and reducing SDS-page gels shown in Supplementary figure 2 have different acrylamide percentages, which makes it impossible to align the gels. Both gels are fully shown.
- Also indicate the units of the values for Maldi.
Please show the graph of MS highlighting the picks for αCD25-m2a and αCD25-m2a-srt-his.
The authors have added the MS graph with unit indication to the manuscript (Figure S2).
Other comments:
- Although it is optional, I would suggest to include a paragraph with the conclusion.
In a final concluding paragraph of the discussion the authors sum up the most important message of the manuscript and have therefore chosen not to include a separate “conclusion” section to the manuscript. However, if the editor prefers to include a "Conclusion" subheading, this can be done preceding the final paragraph of the discussion.
- Along the text, please modify “in vivo” to italic.
The authors have adjusted this throughout the manuscript.
- For the Figure 3, Figure 4 and Supplementary Figure 3: Please indicate the statistical test used and p values legend (as done in Figure 2).
The authors have added this to the figure legends (line 180-181, 206-207).
- Please also indicate for the animal experimentation, how many time the assay was done.
In vivo experiments presented in figure 3 (CD25-m2a treatment synergizes with αPD-L1 treatment) and figure 4 (Continued Treg depletion does not affect therapeutic efficacy) were performed with 8-10 mice per group and performed once. We specified this in the manuscript (line 374).
Reviewer 2 Report
This paper well describes Regulatory T cell depletion using a CRISPR Fc-optimized CD25 antibody. However, there may be minor corrections and supplements necessary for the manuscript. Detailed comments and suggestion are listed below.
Major
I wonder if there are any data on body weight changes in mice during in vivo experiments.
2. The overall outcome data and supporting arguments are great. This paper demonstrated that Treg cell depletion was induced through antibody optimization using CRISPR. Therefore, it is necessary to emphasize the importance of antibody optimization a little more in the discussion..
Minor
1. Need to check the spacing between characters in line 291
2. Need to check the overall unit spacing ex) line 126
3. In the result, the notation of subheadings should be unified.
4. You should check the notation of the unit.(line 193)
Author Response
Major
- I wonder if there are any data on body weight changes in mice during in vivo experiments.
During the experiments the body weight of the animals was not monitored as these animals have growing tumors in their flank. As the volume of the tumor increases, the body weight is not a representative measure for animal welfare. We therefore prefer to use the body conditioning score instead. Treatment with the aCD25-m2a antibodies did not result in a lower body conditioning score for any of the animals. No observations of any aCD25 antibodies negatively impacting body weight or welfare have been made by us or other groups (from literature).
- The overall outcome data and supporting arguments are great. This paper demonstrated that Treg cell depletion was induced through antibody optimization using CRISPR. Therefore, it is necessary to emphasize the importance of antibody optimization a little more in the discussion..
The authors have emphasized this importance in the discussion (line 261-263).
Minor
- Need to check the spacing between characters in line 291
As the text of the manuscript has been aligned to fill the page, some sentences will have a little bit more space between the words by chance.
- Need to check the overall unit spacing ex) line 126
The authors checked the spacing throughout the manuscript.
- In the result, the notation of subheadings should be unified.
The authors have adjusted this throughout the result section of the manuscript.
- You should check the notation of the unit.(line 193)
Unit (mm3) has been checked throughout the manuscript and changed were needed.